# Spreading or Gathering? Can Traditional Knowledge Be a Resource to Tackle Reindeer Diseases Associated with Climate Change?

**DOI:** 10.3390/ijerph17166002

**Published:** 2020-08-18

**Authors:** Jan Åge Riseth, Hans Tømmervik, Morten Tryland

**Affiliations:** 1Department of Social Sciences, NORCE, Rombaksvegen E6 47, N-8517 Narvik, Norway; 2Norwegian Institute for Nature Research–NINA, FRAM—High North Research Centre for Climate and the Environment, P.O. Box 6606 Langnes, N-9296 Tromsø, Norway; hans.tommervik@nina.no; 3Department of Arctic and Marine Biology, UiT The Arctic University of Norway, Framstredet 39, N-9037 Tromsø, Norway; morten.tryland@uit.no

**Keywords:** herders’ traditional knowledge, disease precaution, historical milking grounds, necrobacillosis, climate change challenges

## Abstract

This paper inquires whether reindeer herders’ traditional knowledge (TK) provides a reservoir of precaution and adaptation possibilities that may be relevant to counteract climate change. As our core example, we used the milking of reindeer—which, in some areas, was practiced up until the 1950s–1960s—and the risk of getting foot rot disease (digital necrobacillosis; *slubbo* in North Sámi), caused by the bacterium *Fusobacterium necrophorum*. Via wounds or scratches, the bacterium creates an infection that makes the infected limb swell and, eventually, necrotize. The disease is often mortal in its final stage. Historically, female reindeer were gathered on unfenced milking meadows near herder tents or in small corrals, from early summer onward. When the soil was wet and muddy, the risk of developing digital necrobacillosis was considerable. Our sources included classical Sámi author/herder narratives, ethnographic and veterinary literature, and herder interviews. For this study, we conducted a qualitative review of the literature and carried out individual in-depth interviews with local knowledge holders. Our findings seem consistent: a documented prevention strategy was, in early summer, to move the reindeer to unused grazing land and to avoid staying too long in trampled and dirty grazing land. Contemporary climate change and winter uncertainty due to freeze–thaw cycles and ice-locked pastures challenge this type of strategy. Due to a lack of pasture resources, typical actions today include the increased use of supplementary feeding, which involves more gathering and handling of reindeer, higher animal density, challenging hygienic conditions, and stress, which all contribute to increased risks of contracting and transmitting diseases.

## 1. Introduction

Eurasian tundra reindeer (*Rangifer tarandus tarandus*) have been exploited for food and other subsistence since the last glaciation [1]. In Fennoscandia, large herds of wild reindeer migrated from the coast to inland and were of high importance for all residents [2]. During several centuries, a Sámi hunting and fishing culture was differentiated into several sub-cultures, one of which was reindeer herding. The other sub-cultures comprised inland peasants and coastal fishing peasants [3,4,5]. Archaeological evidence shows that a kind of semi-nomadism or reindeer herding started as early as 500 AD [3]. An early example of domesticated reindeer along the coast of Troms, Northern Norway, was reported at the end of the 800s by the North Norwegian chieftain Ottar (Ohthere) when he visited King Alfred the Great in England in 890 [5,6].

As a result of the intensive hunting of reindeer and the fur trade during the 15th and early 16th centuries, wild reindeer were decimated. As the Sámi had to change their ecological strategies toward reindeer herding and domestication, an intensive reindeer husbandry economy with small herds for meat and milk arose [7,8]. This continued until the early 20th century, when extensive reindeer husbandry with larger herds emerged [9]. The development of reindeer husbandry has been regionally diverse as an outcome of both geography and the effects of nation state policies in Fennoscandia [10].

During the 20th century, reindeer husbandry underwent several transformations. First, there was a shift from subsistence use and milking toward meat and market production. Second, a general modernization occurred, encompassing ordinary schooling for children and family sedentarization, i.e., families changed their dwellings from traditional *goahti* (turf huts) and *lávvo* (herder tents) to ordinary wooden houses like peasants. Generally, this transition was completed during the first half of the 20th century. Third, a change from animal and human muscle power toward increased motorization with snowmobiles and cars commenced in the 1960s. In suitable landscapes, all-terrain vehicles (ATVs) are typically used for person transport, while helicopters are increasingly used for gathering herds, and trucks and barges for animal transportation. Finally, from the 1970s onward, the cooperation with governmental authorities increased along multiple dimensions. In short, this implies significant changes from an independent lifestyle toward a livelihood occupation, increased integration into the broader society, and increased dependency on the state [11,12,13].

Reindeer husbandry industries are also influenced by government regulations, international agreements, institutional barriers, and pressure from the recreation use of the land and loss of animals due to predators. Furthermore, contemporary reindeer herders are, in addition to the effects of natural variability, increasingly feeling the impacts from climate change. Combined, these factors significantly contribute to the direct and indirect loss of grazing land for their animals [14,15,16].

The climate is changing at almost three times a greater rate and magnitude in the Arctic than in the rest of the world, and this is affecting people, animals, and the environment [17,18,19]. During the last half-century, these regions have experienced temperature increases of 0.3–1.0 °C per decade, higher than anywhere else on the Earth, particularly during the winter and spring seasons [17,20].

The Arctic climate system comprises a range of different ecological and physical environments that interact with and feed back to the global climate system, with potential strong influences from changes in permafrost on water and carbon cycling conditions and on the distribution and abundance of ecosystems and species across the Arctic. Earth system models predict that temperatures will continue to increase in the near and long-term future [21,22]. Moreover, the frequency of winter warming events and rain on snow events have increased in the last century and will further increase in the future. For northern Fennoscandia, the prediction indicates a doubling in the number of warming events, while the projected frequencies for the arctic islands of Svalbard are up to three. All of this will increase the climatic stress for the reindeer [23]. A recent report for the high arctic Svalbard archipelago predicts increased temperature (7–10 °C) and precipitation (45–65%), predominantly as rain, as well as other major climate changes [24]. This might indicate a future increase in Fennoscandia as well [17,20].

Climate-sensitive infections (CSIs) can be defined as infections or diseases that are affected by climate-induced changes in the environment, altering epidemiology, geographical distribution, or persistence over time [25]. CSIs represent a new and potentially pertinent challenge to reindeer husbandry [25]. Infectious agents, their survival, and their activity may be directly associated with climatic factors (e.g., humidity, flood, draught, heat, and frost). In 2016, following two unusual warm summer seasons on the Yamal Peninsula (Russia) with the thawing of permafrost, an outbreak of anthrax occurred after the revival of bacterial spores from the ground. Anthrax is a serious zoonosis, and the outbreak killed a boy. In addition, approximately 200,000 reindeer died or were culled during the outbreak [26].

CSIs may also be affected by their hosts’ ability to adapt to climate change (e.g., distribution, population density, and migration), or they may be indirectly associated with climate through the alteration, survival, distribution, or the activity of blood-sucking arthropod vectors (e.g., mosquitos, midges, and ticks) [27]. One example is the filaroid nematode parasite *Setaria tundra*, of which the microfilaria stage is spread by mosquitos (e.g., *Aedes* spp. and *Anopheles* spp.). The development of the parasite to an infective stage in the vector is temperature-dependent, as is also the flying activity of the vector [28]. Modeling has suggested that a summer temperature above 14 °C increases the infection rate, with the disease emerging the following summer if the conditions are still favorable [29]. The results from an investigation aimed at identifying climate-sensitive infectious diseases in the northern regions concluded that arthropod vector-borne diseases (e.g., anaplasmosis and babesiosis) have the potential, in particular, to expand their distribution within the northern latitudes and that tick-borne encephalitis and borreliosis may be classified as climate-sensitive [25].

Many other potential CSIs are affected by extreme weather events, but they could not be clearly classified as climate-sensitive [25]. Still, the predicted increase in temperature and precipitation during summer might increase the occurrences of such diseases toward the year 2100 [24]. 

A broader definition of CSIs may also include some opportunistic infections that could cause disease in immunologically suppressed animals due to climate-induced stress [25]. One example is herpesvirus (cervid herpesvirus 2), which establishes life-long infections and latency in reindeer and can be reactivated, which can thus cause disease outbreaks (e.g., viremia, abortion, and infectious keratoconjunctivitis) following stress [30,31]. Although many infectious agents are associated with climate change, it is often challenging to single out the direct effect of climate change in these often complex pathogen–host–vector systems, since it is often coupled with other factors, such as increased encroachment and loss of pastures, nutritional status, other types of stress, and diseases. However, CSIs represent a new and potentially pertinent challenge to reindeer husbandry, which necessitates mitigations [32].

Reindeer herders have been and still are dependent upon their traditional knowledge (TK) concerning all aspects of their environment, as well as their adaptation skills for their livelihood and survival [33]. This also includes the knowledge and skills in evaluating environmental conditions and clinical signs of the animals to prevent disease by adaptive actions. Traditional knowledge is transferred from generation to generation and is a reservoir for action.

In this contribution, we aimed to explore to which extent reindeer herders’ TK may provide a reservoir of precaution and adaptation possibilities when tackling climate change challenges. To achieve this, we chose to study one specific disease, namely, foot rot (*digital necrobacillosis*), known as *slubbo* in North Sámi, and its relation to reindeer husbandry. 

The primary pathogen that causes necrobacillosis is the bacterium *Fusobacterium necrophorum*, but sometimes, these infections are followed by secondary bacterial infections (e.g., *Trueperella pyogenes* or *Staphylococcus aureus*) [34]. *Fusobacterium necrophorum* is an obligate anaerobic rod and is normally represented in the rumen microbiota of reindeer [35]. Necrobacillosis in reindeer is primarily found in the digits and distal (lower) part of the feet (digital necrobacillosis) (Figure 1), but can also occur in the oral mucosa or other parts of the alimentary tract (oral or alimentary necrobacillosis). The bacterium is not supposed to be able to penetrate intact skin or mucosa, and is regarded as a wound infection, using abrasions or small wounds as the port of entry [34].

Necrobacillosis is rare in wild reindeer [36]; however, there have been several cases in wild reindeer in Norway in recent years [37,38]. Outbreaks in 2007 and 2008 in the Rondane mountains were observed in late summer and autumn, following periods of high precipitation and temperature, which affected more than 80 animals altogether, mostly calves [37]. Similar but smaller outbreaks have also been registered in other wild reindeer populations in Norway [39]. 

However, digital necrobacillosis (which, in some early descriptions, is often referred to as “reindeer foot disease”/“reindeer foot rot”) has long been a well-known and feared disease in semi-domesticated reindeer, especially in the 18th and 19th centuries [40,41,42]. Large disease outbreaks of digital necrobacillosis were often associated with the crowding of animals on wet and muddy ground contaminated with feces. Historically, the gathering of reindeer for milking was a practice that promoted digital necrobacillosis [43]. While the digital form of necrobacillosis practically vanished from the semi-domesticated reindeer in Fennoscandia after the termination of milking, as well as after the switch to larger herds and to extensive herding, the oral form of the disease has been increasingly appearing, associated with new practices of supplementary feeding [34,44].

To date, reindeer herders are still reporting oral necrobacillosis, often recognizing that affected animals lose weight, have problems with handling feedstuff properly, and stop eating, and may have a foul odor coming from the mouth and perforations through the chin. Upon closer inspection, the mucosa of the palate and gingiva may be severely affected with large wounds, and the tongue may be partly or completely detached and fall out [27]. Upon necropsy, mucosal lesions may also be found further down in the gastrointestinal tract (e.g., pharynx, esophagus, and rumen). Suckling calves may transfer the infection to the udder of their mothers [34].

Supplementary feeding has become one mitigation strategy to climate change, when winter pastures have become unavailable due to rain-on-snow events, often forming multiple ice layers on top of nutritious plants such as lichens. The supplementary feeding of reindeer, and especially in corrals and over some time, creating challenging hygienic conditions, increasing animal density and their exposure to infectious agents, such as *F. necrophorum*.

The disease is associated with the warm seasons, but also seems to be promoted by wet conditions [34], which was also reported by reindeer herders in the work of Nensén and Drake [45,46]. Since both the summer temperature and precipitation are predicted to increase toward 2100 [17,19,24], there is a risk that such outbreaks in wild reindeer herds will also increase [38]. However, necrobacillosis is not a typical CSI, since the epidemiology of the causative agent and the disease may not be clearly and directly associated with climatic factors or indirectly via insect vectors. Nevertheless, the appearance of necrobacillosis is associated with herders’ mitigation strategies of climate change, such as supplementary feeding, to avoid starvation and emaciation, animal suffering, and economic loss. Climate change may thus contribute to a renaissance of this historically important reindeer disease, which increases the importance of investigating historical sources and today’s TK on prevention strategies for this disease.

## 2. Methodology

We used veterinary medicine as our background reference, but focused on herders’ TK. We combined several types of sources: Narratives of classical Sámi authors, ethnography, historical veterinary sources, new research, and our own interviews with middle-aged and elderly reindeer herders. We scrutinized the sources of herders’ TK and their practices related to this disease to reveal the historically interconnected dynamics, and we further discussed the challenges for contemporary and future herding.

## 3. Results 

We present our material and findings grouped by sources.

### 3.1. Published by Reindeer Herders

In general, there are not many traditional reindeer herders that have written books. However, we explored several, and most of these books do not discuss or even mention diseases or the prevention and treatment of them. We started with the herders’ narrative, which has the richest description of both this specific disease and its description of herder practice relevant for disease risk management. We commenced onto other sources from this background.

Nils Nilsson Skum (1872–1951) was a Sámi reindeer herder, who was familiar with reindeer herding in the region of Kiruna and Gällivare in Northern Sweden. He was active in the times of herding practice transformation from very intensive to more extensive herding. 

Skum [47] provided an extensive account with consideration of good herding methods from spring to autumn, discussing animal nutrition and wellbeing in relation to available grazing and possible challenges for the sequence of sub-seasons. Particularly interesting is his general recommendation for early summer reindeer herding: first, move the reindeer vertically (up when it is hot, down when it is chilly) in relation to weather (temperature) and insects. *“If it became too hot, it was best to move the reindeer to colder regions with ice and snow”* (ibid., p. 25), also stated by Kjellstrøm [48]. Second, move the reindeer after a couple of days to provide fresh grazing, and more specifically:
*“If it becomes cool weather in weeks and the reindeer have been grazing and trampled around and dirtied to the grazing land, then you should move away to new grass and lichen land. Then you have taken well care of the reindeer”*.(p. 26)

Skum also provided the reason for his advice:
*“With this herding method you protect oneself (e.g., your animals) during hot summer periods against reindeer diseases, which before always used to appear in the start of the hot wave. It was when the reindeer were closer kept together …in the snow free valleys and trampled and whirled up sand dust and tear down their hoofs and legs against sharp stones getting wounds. From that the hoofs becomes swollen and matter is created”*.(p. 26)

Thus far, this is a part of the author’s general advice, not being specific as to whether he had some disease in mind. Furthermore, Skum also delivered parallel advice for rutting time (ultimo September to medio October): “*It was important to move away from dirty and trampled rutting places to a fresh grazing land*” (p. 30).

A contemporaneous Sámi herder who also wrote a book was Anta Pirak (1873–1951) from Jokkmokk, Northern Sweden [49]. His book is an account of all sides of the life of the Jokkmokk Sámi, but it has only one paragraph about disease:
*“There are many diseases that reduces the number of the reindeer. The disease that **most often** hit the reindeer is the hoof disease* (foot rot disease). (The preface substantiates that Pirak at least knew the conditions for reindeer husbandry in the Jokkmokk area very well, and accordingly, his statement has a more general relevance than his mere personal experience). *“The Sámi divides the foot rot in two types. The one type recedes, but the other type does not recede, on the contrary the reindeer die, regardless of how strong. We call it “the Club””* (*slubbo*) (The Sámi name slubbo literally means the club, as an infected foot often resembled the shape of a club (Figure 1)), *“while the hoof of the reindeer swells, it dies. When the foot rot disease hits a reindeer, the best medicine is to let the herd free, without herding, so it can spread out. If you drive such a herd collected, or let lay down on the resting place, then the disease hits healthy reindeer”*.[49]

Though short, the citation points out (1) foot rot as the most common disease and that (2) the best treatment was to let the herd spread. We should keep in mind that these narratives stem from the time before any medical treatment existed, i.e., no antibiotics as we have today. Furthermore, one important point when letting the herd spread is that (depending on the disease/infectious agent) the disease may be transmitted/introduced to other herds. Slubbo is probably associated with environmental conditions, and the pathogen is carried in the rumen, but an agent such as foot and mouth disease virus would be devastating if the infected herd was spread out, as well as, e.g., chronic wasting disease (CWD)/prions. 

Returning to Skum’s account; later in his book [47] he provided a very detailed and concise description of the foot rot disease:
*“In the shedding month* (June-July) *the hoof disease* (foot rot disease) *starts, especially when there are wounds or scratches on the hoofs or at the sides of the legs. It becomes abscesses, with fluid and matter in the legs, which swell. It does not recede without creation of matter. Sometimes a stark foot rot epidemic breaks out, which infects other reindeer coming to the pastures where these foot rot infected reindeer have grazed and wandered about”*.(p. 42)

Skum also provided a more detailed description of how the disease can develop:
*“It has also happened,……, that the abscess starts at a healthy hoof without any wounds. Usually it starts at the back hoofs. If it only strikes the front hoofs, it usually blows over faster. If it starts in the back legs, usually the gland between the thighs”* (what is “the gland between the thighs”? Is he maybe talking about lymph nodes? The inguinal lymph nodes are probably the ones, if this is the case (but a lymph node is no longer called a gland)) *“swells and the swelling can become as large as a medium sized potato. When if the swelling disappears from there, it may settle in the back hock and the knee joint and may also create open wounds in the thighs*.Of cause this itch, and the reindeer rubs it with its nose or its lower mandible teeth to end the itching. This way the foot rot has become so persistent that permanent wounds in the thigh and shoulder areas and the shoulder gland that also have become influenced of the disease. And the last phase of the disease develops to what is called sis-ruod’no”(known as a complication of *slubbo* [39])
*“which starts when the reindeer licks the permanent wounds. Later other reindeer get the disease through the mouth, when they eat from ground where these animals have eaten. And then a reindeer pest”* (reindeer pest (*Pestis tarandi*) is a very serious but seldom seen disease. The last known case in Sweden took place in Jokkmokk in 1896. It is most likely that Skum used the word in a more general sense, meaning an epizootic) *“starts, and immense damage can evolve”*.(pp. 42–43)

We note that the author provided a very accurate case history. Furthermore, Skum also offered clear treatment advice:
*“The first you then must do, is to move to a new area where the land is healthy. The best is to drift the reindeer down to the forest land to an area with wet marshes. There they can rinse the wounds, there are twigs and leaves, on which they can rub away the crusts of their wounds and get them clean. Insofar as the Elders have told, this have been the best cure, and this has been practiced from times immemorial”*.(p. 43)

We note that Skum’s advice was not only to move away from the disease, but also to move down from the mountains to wetter areas where the animals could rinse their wounds by rubbing.

### 3.2. Scientific Publications

Several sources of traditional knowledge are put together in Table 1. Most of the table is based on sources collected by the language and culture researcher Just Qvigstad [50]. (https://nbl.snl.no/Just_Qvigstad).

First, a short notice about the core sources. Professor Carl von Linné (1707–1778) (https://no.wikipedia.org/wiki/Carl_von_Linn%C3%A9) is best known as the “father of modern taxonomy,” but, importantly for our topic, in 1732, he went on a six month expedition to Sápmi, which became the basis for his *Flora Lapponica,* also including extensive descriptions of Sámi life and use of nature. By that, he is also considered as important for modern ethnobiology [51]. Approximately half a decade earlier, Johan Scheffer provided a description in his work: “Histoire de la Laponie” about “*the great nuisance inflicted upon Lapps and reindeer by an **espece de grans moucherons**.*” He described how the reindeer are moved to the highest mountains in order to get rid of pests such as mosquitos, midges, and flies, and how the Sámi drive away the bloodsuckers with fire [52].

Lindahl and Öhrling’s *Lexicon Lapponicum* [53] was one of the first Sámi dictionaries. Johan Turi is known as the first Sámi that published a book, namely, *Muitalus sámiid birra* (Stories about the Sámi) [54], which is an extensive narrative of the Sámi natural and spiritual world. He lived in the Torneträsk area of northern Sweden and was an experienced reindeer herder before he settled down as an author. Jonas Nensén was a vicar in Västerbotten, Sweden, that collected folk culture. His collection [45] was the basis of Sigrid Drake’s dissertation “Västerbottenlapparna” (The Sámi of Västerbotten, the second northernmost county of Sweden) [46]. Several of the other sources are priests and veterinarians.

From Table 1, we can sum up that already from the 18th century, the descriptions of the disease, its cause, and its development are relatively uniform and in line with both Skum’s description and veterinary science. The only deviation is Linné [55] (Linné wrote about this disease already in 1732 [43]), who stated the warble fly as a possible cause. This explanation is also cited by two other sources, which both present it as “old people say,” but also provide a “modern” explanation as their own view, e.g., Johan Turi [54], indicating that this was an old explanation among the Sámi. There are also other diseases that have been blamed on the warble fly, e.g., infectious keratoconjunctivitis (IKC). Animals with slubbo might have gotten more warble flies, maybe since they move slow and are less fit, reducing their capability to escape egg-laying flies.

Already, the second source [53] understands the disease as a contagious infection. This understanding also has implications for the prevention and treatment, both for the already infected animals and for other animals and herds. For the individual reindeer, the case is mainly to prevent licking by covering the wounds by oil boiled on bark, tar, or turpentine, and even nitric acid, as well as to physically rinse the wounds. To prevent the spread of the disease, the reindeer should be kept apart and moved to healthy pastures. The connection to milking practice seemed to be understood by the late 19th century, as well as the possible contamination of walking in the tracks of infected animals. Approximately one century ago, the connection to muddy ground also seemed to become clearly understood. We can sum up that in the early 20th century, the sources substantiate that an understanding in line with Skum’s seemed to be widespread among Sámi reindeer herders.

We will come back to some of the other sources in the following section.

### 3.3. Infectious Diseases and the Development History of Reindeer Husbandry

Infectious diseases have obviously played a more important role in reindeer husbandry historically than in recent times. Furthermore, they also seem to have been one of the major factors influencing the development of reindeer husbandry. The others are government policies, climate variability, physical encroachments, and other external pressure [10,60,61]. However, the importance of the historical role of infectious diseases has still not been fully explored, but there are clear indications that it is considerable.

The Sámi language professor Israel Ruong provided an important contribution to the regional history of northernmost Sweden and the adjacent areas in Norway. Ruong pointed out that from the mid-18th century, there were large outbreaks of infectious diseases in reindeer herds in northern Sweden, with very large losses of reindeer and (more or less) the collapse of entire reindeer herding communities. At the end of the 18th century, the population of Torne Lappmark (Sweden) was reduced from about 1300 to 800 people, and many reindeer herders left reindeer husbandry and settled in Norwegian fjords where they lived as peasants with small farms combined with fisheries [62]. This story and its wider context should be further explored, as it is an important aspect of Sámi history and the local and regional history of large parts of northern Scandinavia.

Here, we use Ruong’s [62] reasoning to sketch what seems to be the main relationship between infectious diseases and reindeer husbandry development. One of Ruong’s important theses was that *“The type of reindeer husbandry is a function of the landscape,”* i.e., that the landscape form determines which reindeer husbandry type is possible. Sámi terms such as *oaggás eatnan* (landscape with natural borders) and *luomokis eatnan* (landscape without natural borders) (op. cit., p. 69), as well as *octilaš eatnamat* (landscape that collects the reindeer), support this. Ruong’s analyses are a good starting point for studying the relationship between reindeer husbandry and disease.

One example is Ruong’s [63] statement about herder opinions in Jukkasjärvi (includes the Sámi villages Saarivuoma, Talma, Rautasvuoma (now Gabna), and Kaalasvuoma (now Laevas) in northernmost Sweden). We then need to know that the situation for reindeer husbandry in this region during the late 19th and early 20th centuries was rather chaotic as a wider consequence of the closing of the national borders between Norway–Finland (1852) and Sweden–Finland (1889). Due to a major immigration of herders from Kautokeino, both the pressure on and the competition over pastures increased dramatically [10,63].
*“Reindeer diseases are often connected to an all too far domestication of the reindeer herds* [Several sources among them Turi [54]] *hold that more free herding among the North Sámi*” (From Kautokeino, Finnmark, Norway (our comment) [10]) “*has decreased the reindeer diseases in Jukkasjärvi. This opinion is now firmly based among the Sámi in Jukkasjärvi*” [63] (p. 20).

This statement is very interesting and relevant to the feeding boom we experience today, and is also in line with veterinary sources from the shift between the 19th and 20th centuries [42,43,44], which substantiate that professional authorities clearly saw the connection between infectious disease(s) and herding practice. Bergman [42] stated as a general picture for Sweden:
*”In Sweden it is generally acknowledged that the infection spread more easily with the intensive reindeer herding, when the reindeer during summer are kept under continuous guarding on a restricted room in reindeer corrals for milking”*.(p. 29)

Ruong realized that milking-type reindeer husbandry meant a double risk. First, daily roundups of the herd—if the milking places were not in well drained and dry areas—were a source of disease transmission. Second, milking and hard grazing reduced the uptake of food, especially during the first part of the summer as the grazing range of the reindeer is limited to higher lying areas due to disturbances from mosquitoes, warble flies, and heat. The outcome was nutritional deficiencies among both females and calves. In particular, the calves became leaner when access to milk was restricted. This made the calves especially prone to infectious diseases and probably less fit to survive the first winter.

Ruong [62] therefore concluded that the risk of infectious diseases put a limit on how hard one could tend and increase the degree of tameness in reindeer flocks (p. 68). Exploring historical changes in reindeer husbandry, he found that reindeer herders therefore began to allow the reindeer to spread to limit the risk of infectious diseases. This meant a gradual transition to summer-extensive reindeer husbandry, i.e., less herding, with larger flocks and less intensive land use.

Typically, in the largescale landscapes of Finnmark, reindeer herders already in the late 1700s and early 1800s brought reindeer to the peninsula and the islands, where the reindeer were (more or less) allowed to spread out and graze freely. Such summer-extensive operations prevented any infectious diseases from flourishing and spreading [56,57]. This mode of operation was spread to most of Sápmi as a result of border closures and other government regulations [10] from the latter half of the 19th century.

Leaving Ruong’s explanation, we turn to regions with landscapes suitable for milking-type reindeer husbandry, where movement along an altitudinal gradient was important. The first known description of historical milking grounds (Renvall [46] in Swedish—the translation “milking ground” [64] is more inclusive than the alternative concept “reindeer pen” [65], as milking in many cases was performed without the assistance of any fence) is from 1671, when governmental officials were sent out to map Umeå lappmark. In his description, the clerk Anders O. Holm wrote:
*“around the Sámi hut or house, where he has had his animals during summer, was a beautiful meadow grass, lush and long, so one could wonder”*.[66] (p. 25)

This “*beautiful meadow grass*” is created when reindeer are collected and kept within a pen, partly for milking and partly for protection against insect harassment with smoke from an open fire. The reindeer trample and fertilize the ground, which becomes very nutritious. This site (Åskilje https://www.google.com/maps/place/920+51+%C3%85skilje,+Sverige/@64.7475891,15.6379299,8z/data=!4m5!3m4!1s0x46798cf7550a00e5:0x759b77a219e0d0db!8m2!3d64.916667!4d17.8666669) is in the Västerbotten taiga, and about forest Sámi (having only short local migrations during their annual cycle), but Holm also described reindeer herding from a location (Byrgfjäll/Björkfjället, near Ammarnäs, Västerbotten) of more than 1000 m above sea level (op. cit., 25–26).

The forest line (*orda* in North Sámi) has an important role in Sámi landscapes, historically both for the hunting of wild reindeer and traditional reindeer husbandry, as well as for modern reindeer husbandry, e.g., both rutting places and calving areas are usually located there. Spring and autumn dwelling sites were places just below the tree line, as they were in shelter but were also strategic [62].
*“Before when the mountain Sámi milked the female reindeer regularly, the location of the dwelling site close to the tree line was important as on sunny days one collected the flocks on the bare mountains and took the herd to the milking ground by the **goahti** (dwelling place) for milking”*.(p. 14)

There are several descriptions of the historical milking grounds and their locations:
*“The milking was often performed on naturally delimited locations as elevated plateaus and headlands or in pens of stone or trees and twigs”*.[48] (p. 84)
*“The animals were collected in pens or on headlands and other suitable places without any need of fences”*.[67]

We note that the milking grounds were deliberately chosen to be suitable locations, with or without fences. Ruong placed them just below the forest line, while Fjellström [68] wrote that they were in the mountains above the forest line. Vorren [69] also reported milking grounds in the mountains of Helgeland that occurred hundreds of years ago, and Kjellström [48] also illustrated a picture of milking on a snow bed in the mountains.

The location of the milking place changed during the season related to the grazing area. Following the account of Kristoffer Sjulsson [70], the milking pen was made ready during the week before summer solstice:
*“The milking ground was usually placed upon a dry heap some distance from the tent location, which had some large birches. The pen was made from birch trunks and twigs”*.(p. 243)

Furthermore, in summer, the reindeer were collected in the high mountains for milking. They were collected on a spit on a mountain lake. If it was particularly hot, the milking took place on a snowdrift [64].
*“The milking took place once or twice every day. At the milking grounds the soil soon were grazed down and trampled. Due to the danger of pasture deficit …. And to avoid diseases as the hoof disease [digital necrobacillosis] milking ground were shifted many times during the milking season“*.[48] (p. 84)

Sigrid Drake provided important evidence about traditional reindeer milking practice and site management during the late 1800s and early 1900s [46]. These Sámi also came into adjacent parts of Norway at Helgeland (Byrkije (Børgefjell), Hattfjelldal, and Rana). The milking grounds were primarily used for milking, but also as calf marking sites and for other activities. Some Sámi had large herds and used their milking grounds carefully and only for a few days. Smaller herds could use them for two or three weeks. The Sámi had a distinct terminology covering a variety of milking sites of different ages, conditions, and locations.
*“They had many milking grounds, even up to 30 or 50. They stayed only 3 or 4 days, sometimes only overnight, at one field, as the reindeer managed better when not trampled at the same spot and not destroyed the pasture. Tracks between lakes and fields were marked on trees. The fields were cleaned from windfallen trees etc. [as well as branches]. When much rain, they needed to change after two weeks because of faeces accumulation. Next year this was dried so that the same field could be used……Grounds with **jåmo**” (**jupma**) (Rumex)* (The Sámi made a porridge of *Rumex acetosa* boiled in reindeer milk, fermented and stored in reindeer stomachs for winter food) *“could not be used before the second year as the roots will be trampled”*.[46] (p. 62)

Furthermore, Drake stated about milking grounds that: “Good Sámi walk far for milking, they do not herd” (the reindeer) to the dwelling place as it could be unclean” (ibid., p. 61).

The Byrkije (Børgefjell) area on the border between Norway and Sweden, which formed part of Drake’s study area, might be the area that is mentioned by Nensén [45], since the Swedish part of Byrkije is included in Åsele lappmark. The milking of reindeer in Byrkije ceased completely in the 1920s–1930s, probably as a result of larger outbreaks of diseases such pasteurellosis in Frostviken in 1924 in the same area [71,72].

What is remarkable is that old milking ground locations, due to fertilizing by reindeer feces, urine, and trampling, are still green 80–90 years later (Figure 2). Originally, poor vegetation became “greener” due to intensive use, and more than 10% of the landscape (the total is over 1000 km^2^) is related to milking-type reindeer husbandry alone (cultural remains of tent settlements on and near milking pen locations bear witness to this), while over 50% of the vegetation is more or less affected by reindeer and reindeer herding [65]. These results have been confirmed by laboratory experiments, substantiating the role of nitrogen addition to poor vegetation [64,65].

### 3.4. Preliminary Summary

Summing up our information from written sources, we note that the writings of the two herders in the 1930s describe the disease broadly in accordance with veterinary knowledge. That is also the case with most of the descriptions and explanations in Table 1. Both Skum [47] and Pirak [49] have as their main advice to let the herd spread and move to new land. The prevention strategies in Table 1 are also in line with this. What is more questionable is whether any of the listed treatment suggestions help. The section of how diseases have influenced the development of reindeer husbandry suggest that the practice of reindeer herding changed because of the adaption to the risk of disease, either to a more extensive type without milking or to establishing many milking sites, preferably dry ones and changing between them often. In the next section, we receive an indication of how much of this TK has been transferred to old and middle-aged reindeer herders. 

### 3.5. Interviews

In northernmost Sweden, Interviewee 1 (born 1944) knew about herding practice in the 1930s due to family narratives. Calf marking was performed on snow patches in the mountains (like the strategy mentioned by Drake [46]) or on a small peninsula or blocked (by a rope watched by dogs) area between ponds or small lakes. They aimed to avoid reindeer visiting that same spots several times each summer, as they had heard that the reindeer could be contaminated if they trampled in one place too long. The herders clearly knew this and took precautionary measures to avoid such contamination.

Another form of milk-based reindeer herding occurred in the Vesterålen Islands, in the northernmost west of Nordland County of Norway, from the mid-1800s up until at least World War II. Sámi farmers still kept both reindeer herds and livestock. Interviewee 2, born 1947, remembered this well, having been a child in a herder family. The last milk-based reindeer herding practice took place in the late 1950s.

For our purpose, the most important of the memories of Interviewee 2 were that there were no incidents of slubbo/necrobacillosis or other serious infections in his region, and perhaps most importantly, he provided a clear explanation of why—which was clearly related to the herding and milking practice. The herders had relatively small herds, and as milking sites, they used dry hills and rotated the milking site every third week. Furthermore, they chose a new series of milking sites every year and instead used last year’s milking sites for growing potatoes. It seems quite clear that the reindeer herders integrated a very high consciousness of infection risk into their practice.

Our last interview at Hinnøya Island, also in Vesterålen, is a quite modern example. A family company, with a history from 2008, is run by wife and husband reindeer herders. Close to cruise ship routes, it has gradually grown to become a major regional tourist destination. Tourists meet reindeer in a corral and catch some glimpses of Sámi culture through food and amusing narratives. The corralled reindeer are a limited part of the total herd, as the majority are free grazing. Interviewee 3 (born 1957) reflected over keeping reindeer in corrals for a long period of time. He told us that reindeer get swollen hooves when they stay in a corral for a long time (i.e., months). Then herders plan to change the corral site for precautionary reasons, and they keep the animals in extra circulation; e.g., they keep them in “quarantine” in a special corral before they can reenter the main herd. In addition, they also circulate the few very tame animals.

This is a conscious strategy to prevent disease. Interviewee 3 justified this strategy as an alternative to being super restrictive, which would be demanding in many respects, and provided his main arguments. First, the extra income from tourism reduces the need for having many reindeer and, accordingly, reduces the pressure on land. Second, the prevention of disease is based on intuition, which again is based on experience. Our reflection on this is that it is a clear sign that old learnings are integrated into the Sámi reindeer herder culture and are still alive, brought forward and used whenever needed.

## 4. Discussion

In the written reports of reindeer herdsmen, ethnographic literature, and the interviews we conducted, it is stated that reindeer herdsmen have historical experience that keeping reindeer herds too close and too long in the same place increases the risk of disease. This seems to have established itself as part of a (more or less) collective memory, i.e., a part of Sami traditional knowledge that forms part of the basis of reindeer husbandry practices or as an element of good reindeer husbandry practice [46,63]. For example, the frequent rotation of milking grounds mentioned by Svonni was also reported by Drake [46], and if the herd was large, use of the same milking pen for 3–4 days was the limit. Even for small herds and after heavy rainstorms, they had to shift the milking site and clean the used pen very soon and before 14 days (ibid.). Considered as good reindeer herders were those people who milked the reindeer far away from the dwelling site, since the latter could be contaminated by faeces (ibid.). Anyone who knows and recognizes the practice does not need to know the background.

Modern reindeer herding is based on larger herds than historically used. Since the end of the 1960s, reindeer herding in Sápmi has also become increasingly based on the use of various vehicles. However, the landscape, as per Ruong’s distinction, determines which vehicles can be used. Many districts therefore also have a higher degree of tameness than usual, at least for parts of the flock, in order to perform migration through demanding landscapes. Modern communication technology with telemetry, Global Positioning System (GPS) devices, and drones is also increasingly used.

Despite all the technology use, it is important to be aware that reindeer herding is still based on a comprehensive knowledge of nature and the landscape, social relationships, reindeer and their habitat, and response patterns in various situations. This knowledge is still transmitted between generations—note, e.g., that Interviewee 1 knew well the old family narratives from before, while at the same time was renewed with the young people’s own learning and experience building.

It is also worth noting that many districts in recent decades have undergone changes in work with the herd, causing less stress on the animals, such as calf marking without the use of lasso (*suopan*) but instead either through the use of binoculars in a special corral or a traditional method with collection on a snow patch. Both methods contribute to a greater degree of tameness.

Reindeer herding in Sápmi is currently subjected to extensive stresses due to increasing predator strains and nature interventions, both as a result of infrastructure development and new recreational installations; i.e., both tourist facilities and private cabins, as well as direct disturbances from other nature users such as hunters and hikers [73]. This occurs in parallel with climate change, which leads to unstable winter grazing and unsafe migration routes in both the spring and the autumn [74].

While the most immediate reaction of reindeer to decreased access to winter forage is to spread, reindeer herders often act by increasing control, namely, moving the herd or providing supplementary feeding [75]. Supplementary feeding can be a mitigating action both to pasture fragmentation and ice-locked winter pastures [23,76]. This practice has gradually developed in Fennoscandia since the late 1960s, with Finland as a forerunner, Sweden the first to follow, and Norway somewhat slower, introduced to increase the winter survival of reindeer under critical conditions. The late winter and spring 2020 in Finnmark County, Norway, reached a massive scope to avoid starvation.

However, it has become more and more common to establish feeding regimes, as a supplement or as full ration feeding during several months of the winter [77]. The successful feeding of reindeer is dependent on knowledge, experience, and high-quality feedstuff, and a wide range of diseases directly associated with feeding have been documented, such as ruminal acidosis, diarrhea, and bloat [78]. In addition, feeding, especially over longer periods of time and in corrals, increases animal density and challenges reindeer with unfavorable hygienic conditions, quite similar to the experience and traditional knowledge referred to above with regard to slubbo and its association with milking practice [34,79]. Several outbreaks of IKC, contagious ecthyma, and alimentary necrobacillosis have been documented during the past few years in reindeer herds on different feeding regimes in corrals, sometimes affecting several dozens of animals [44,80].

The encroachment of reindeer pastures, climate change, and the increasing feeding practice represents a major dilemma for today’s reindeer herders [77]. Mitigating actions often mean the increased gathering and handling of reindeer, which means that the reindeer are exposed to more stress, higher animal density, challenging hygienic conditions, and infections in new environments. This contradicts the traditional practice of avoiding keeping reindeer too tight. Herders from Finland, Sweden, and Norway also express negative feelings about feeding, as it is not a free or wanted choice (ibid.). However, the degree to which such attitudes are representative remains to be explored.

At the same time, climate change and increased temperatures increase the risk of infections and new diseases [81]. Another part of this dilemma is formed by cultural factors, e.g., reindeer do not belong in fences, their meat may lose that reindeer taste, and feeding necessitates investments in pellet containers, which can be seen as gradually changing from herding toward farming [82]. Furthermore, the old mitigation strategy mentioned by most of the sources above—i.e., to move the animals to alternative pastures—has become much more difficult due to the fragmentation and loss of pastures, which is also probably associated with increased feeding.

## 5. Conclusions

We started out by asking if the historical and traditional way of managing the disease digital necrobacillosis, also known as *slubbo*, in reindeer herding may provide insight into the potential that traditional reindeer herder knowledge has in the ongoing encounter of climate change-related challenges. Our main finding is that traditional knowledge of the mitigation of infectious diseases, such as necrobacillosis, refers to moving animals to alternative pastures or to spread the reindeer (from intensive to extensive reindeer herding) and to decrease animal density and stress. However, current multi-stressor pressure, such as pasture encroachment and climate change, is increasingly forcing herders to conduct supplementary feeding on a regular basis, putting reindeer into a similar situation as that in which the milking animals were kept, possibly resulting in subsequent diseases, such as *slubbo*. This puts reindeer herders in a difficult situation and challenges both the traditional knowledge and the cultural feelings regarding what reindeer husbandry is all about. 

## Figures and Tables

**Figure 1 ijerph-17-06002-f001:**
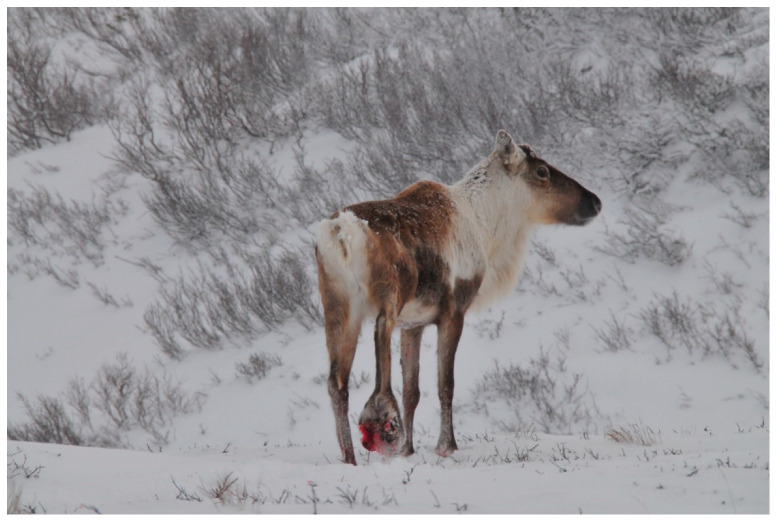
An adult male wild Eurasian tundra reindeer (Norway, January 2015) with digital necrobacillosis (*slubbo* in North Sámi). The distal part of the right hind leg was extremely swollen and was bleeding. The condition was very painful, and the animal only reluctantly used the hindleg. The bull was shot for animal welfare reasons, and pathological findings (Norwegian Veterinary Institute) confirmed the diagnosis. Photo: Erik M. Ydse, Norwegian Nature Inspectorate.

**Figure 2 ijerph-17-06002-f002:**
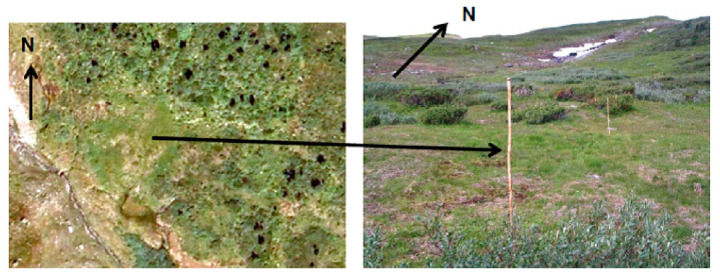
Reindeer milking landscape (buhtjeme-aevieh) in Byrkije 80–90 years after milking practice ceased. Aerial image (Statens kartverk) on the left and ground picture taken by Sigbjørn Dunfjeld to the right [65].

**Table 1 ijerph-17-06002-t001:** Traditional knowledge on foot rot (digital necrobacillosis) in the literature.

	Description and Explanation	Prevention	Treatment	Original Source [Our Ref.]	Liter. [Our Ref.]
1	The feet swell at the margins of the hoofs and rot, so that the animal limps and cannot follow the herd; this disease emerges especially in strong sunshine and may be caused by the warble fly.			Linné [55]	Qvigstad [50], p. 6
2	Reindeer often suffer from this disease in the summer.	It is very contagious, so the Sámi look after their reindeer very carefully so that they do not come close to infected animals; by that, they can at least prevent tongue and mouth infection by licking, because if contracted this way, the animals always die.	They usually cure this disease by applying lubricating tar to the wound and sprinkling gunpowder upon it.	Lindahl and Öhrling [53]	Qvigstad [50], p. 6
3	Starts with swelling between the hoofs and the creation of matter before finally, the swelling opens and the matter run out; always starts in the summer and ends toward the winter, culminating in the feet rotting.	Seen as the most contagious, but the most uncertain.		Grape [56]	Qvigstad [50]. p. 7
4			Turpentine oil greased on the wound to prevent licking.	Sidenbladh [57]	Qvigstad [50], p. 13
5	Before, many mountain Sámi milked their reindeer the whole summer and forced them into a small corral where they had to stay for hours, which caused the accumulation of feces that, according to the Sámi, brought forth the disease.			Vickar G. Balke, Karasjok, 1880–1885	Qvigstad [50], p. 9
6	The Sámi tell that healthy animals are contaminated by going into the tracks of sick animals.			Horne [42,43]	Qvigstad [50], pp. 9–10
7ab	In old days, they milked a lot in the summer, resulting in the breakout of all kinds of reindeer diseases, which killed many reindeer due to their highly contagious nature; such diseases are also contagious via footprints because of the swelling between the hoofs resulting in the excretion of matter, sometimes lasting the whole winter, but not necessarily the summer, with the contamination of only one or two reindeer as a result of jumping insects on warm summer days (old people say then it becomes *slubbo*).		Lubricating all sick tissue with bark oil as thick as tar.	Turi [54], p. 31 Turi [54], p. 35–36	Turi [54], p. 31 Qvigstad [50], pp. 7–8
8	The disease is rare in Varanger; the Sámi hold that the reason is that animals roam freely in summer; unlike in Karasjok and Kautokeino, where the animals are herded and often held tight together.			Reg. vet. H. Olsen Veterinærvesenet, 1903	Qvigstad [50], pp. 10–12
9		Where an infected reindeer has trampled down the grass; when a reindeer limps, it should be checked.	Lubricating with thin tar, mixed with cod liver oil, so the reindeer do not lick to avoid getting the disease internally, otherwise pneumonia and rapid death will follow.	Nensén (Åsele) [45]	Drake [46]
10	The disease appears to be due to the long stay of reindeer on milking ground, so it gets too muddy; lonely reindeer do not get this disease.	The old Sámi would not let reindeer stand in ponds or lakes of water, otherwise they would ruin their feet on stones and develop club disease; preferably, they should stand on “tsoevtse”—firn (icy snow) in mountains—and there should be frequent changing of milking grounds away from the dwelling site.		Nensén (Jokkmokk and Åsele) 45]	Qvigstad [50] Drake [46] Drake [46]
11			The best means to protect against club disease is nitric acid.	Drake [46], p. 258	Qvigstad [50], p. 7
12			Applying oil from willow bark as thick as tar, mixed with salt.	Isak Eira, Kautokeino	Qvigstad [50], p. 8
13			Applying turpentine oil or tar onto the wound.	Anders Eira, Kautokeino	Qvigstad [50], p. 8
14	The old Sámi say that the most dangerous type of foot rot is caused by insects; this is only a belief, and it is a contagious infection.			Forester Gløersen, Karasjok Precentor O.Hagen, Karasjok	Qvigstad [50], p. 10 Qvigstad [50], p. 10
15			Greasing rowan bark boiled in urine on rotten wounds on the feet.	Smith [58], p. 367	Steen [59], p. 11

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
