# Peer review of "Spreading or Gathering? Can Traditional Knowledge Be a Resource to Tackle Reindeer Diseases Associated with Climate Change?"

_ijerph, 2020, doi:10.3390/ijerph17166002_

Round 1
Reviewer 1 Report
This is a really interesting paper on a pertinent subject. The work is clearly presented and the English language is mostly good. However, there are some minor grammatical and stylistic corrections to be made – preferably by an English speaker. These are quite minimal and are mostly translational matters. Overall though, this is a good and useful contribution to the subject.
Author Response
Response to Reviewer 1 Comments
Reviewer 1 only commented that there is a need for minor grammatical and stylistic corrections to be made – preferably by an English speaker.
Response
We will send the manuscript to English editing after final approval of the content.
Reviewer 2 Report
The authors present an interesting study on reindeer herder’s traditional knowledge (TK) and how they face off the challenges provided by the impacts of climate change. Despite they focus on the fight against a digital disease (necrobacillosis) and maybe the title do not adjust with the development of the paper I believe the paper is interesting and fits to the scope of the journal.
Anyway, in order to get a definitive approval of the manuscript, I recommend some recommendations:
Title:
I will recommend changing it to a more concise description, adjusting it to what the reader is going to find.
Abstract:
It could improve. It should contain an explanation of the Fusobacterioum effects (short). I will take out the “documented prevention strategy”.
Introduction:
- The prase “Sámi reindeer herding developed from a hunting and fishing culture” is difficult to understad, could you explain better ?
- I dind’t use footnote to explain in english the terms “goahti” and “lávvo” merely I add the translation just after the concepts.
- The same for “lower part of the feet” (footnote 3).
Material and methods:
- The same comment than above for footnotes.
- The paragraph regarding the history of Nils Nilsson Skum, I think that is not necessary.
- I will try to summarize the two paragraphs regarding Skum recommendations and do not use his exactly words. Just a synthetic paragraph of his inputs.
- The same for the comments of Anka Pirat.
- After reading completely the Basic Sources and Infectious diseases and the development history of reindeer husbandry sections I strongly recommend summarizing the information in few paragraphs due to the information is showed in the interesting table 1.
- I will add a new section for the interviews. Additionally, I will try to be more concise and summarize the history of each surveyed person.
Results:
- There are no results due part of them are in Material and methods block. So, I recommend separate methodology and results in order to the paper flows more consistently.
Discussion:
- I will use the concept of interviewed instead of the name of the person.
- I’m not sure if using “op. cit.” is correct in this journal. Please, check it carefully.
Conclusions:
Some sentence related to the future of the herders and the potential new diseases inducted by the climate change is needed. Maybe also some recommendation about the importance of TK.
Author Response
Response to Reviewer 2 Comments
Reviewer 2 had several suggestions.
Title: to a more concise description, adjusting it to what the reader is going to find.
Response: We have adjusted the title to: “Spreading or gathering? Can Traditional Knowledge (TK) encounter for Reindeer Diseases associated with Climate Change?”
Abstract: a) a short explanation of the Fusobacterium effects. b) take out the “documented prevention strategy”
Response: a) We have provided a short explanation b) We disagree. We hold that the prevention strategy is important and a core issue of the manuscript. It is also reflected in the title.
Introduction: 1) How Sámi reindeer herding developed from a hunting and fishing culture should be explained better.
Response: We have rephrased the sentence and connected it better the section it is an introduction 2.
2&3) Changing from footnotes to text.
Response: We have followed the advice and there are not any footnotes left.
Material and Methods
1) Changing from footnotes to text.
Response: We have followed the advice and there are not any footnotes left in this block.
2) Removal of paragraph about NN Skum. Response: Done
3) Only summarizing, not citing Skum. Response: We disagree. Skum is a core source, and we find it very interesting to cite his input thoroughly to provide an accurate story for comparing with veterinary sources.
4) Removal of paragraph about A Pirak. Response: Done
5) Provide a new summary. Response: Done
6) Add a new section for interviews and be more concise. Response: Done
Results
1)Lack of coherence between blocks Response: Changes made in order to get a better flow.
Discussion
1) Use interviewed instead of name on person. Response: Done
2) Can op. cit. be used? Response: Changed to ibid. which is allowed
Conclusions:
Some sentence related to the future of the herders and the potential new diseases inducted by the climate change is needed. Maybe also some recommendation about the importance of TK.
Response: we find this unnecessary, as it would only be repletion of what is already said.
Reviewer 3 Report
Thanks for the opportunity to revise the manuscript “Spreading or gathering? Is Traditional Reindeer Herding Knowledge Relevant When Encountering Climate Change Challenges”. This paper inquires if reindeer herder´s traditional knowledge (TK) provides a reservoir of precaution and adaptation possibilities which may be relevant to counteract climate change. Despite an interesting topic, the article fails to relate the studied disease to climate change. For example, the relationship between diseases in general and climate change could be much more explored in the introduction, which would provide a greater basis for the study.
In addition, further explanation of Necrobacillosis is also needed. What studies exist on this disease, and what is the relationship between climatic factors and its incidence? As it is, it is vague and still indicates that mismanagement can be more significant than climate change. Furthermore, the way the study was conducted does not apply to the scope of the journal. I believe that the format of the article fits more in a social science magazine.
Author Response
Response to Reviewer 3 Comments
Comment 1.
the article fails to relate the studied disease to climate change. ….., the relationship between diseases in general and climate change could be much more explored in the introduction,
Response: We have included a more in-depth definition of CSIs, including some examples of diseases (host-pathogen-vector) that may be affected by climatic factors. The association between infectious diseases and climatic factors are often vague, as it is often difficult to single out the effect of climatic factors. This has also been commented in the manuscript.
Comment 2
In addition, further explanation of Necrobacillosis is also needed. What studies exist on this disease, and what is the relationship between climatic factors and its incidence? As it is, it is vague and still indicates that mismanagement can be more significant than climate change.
Response: As for necrobacillosis, we have mostly older descriptions of outbreaks, from a time when climate change was not on top of people’s mind, maybe. Outbreaks have been associated with the warm season, as well as with wet conditions, but there are hardly data to support or quantify these observations. As seen today, we have observed necrobacillosis in wild reindeer (digital form) during wet conditions, and also in semi-domesticated reindeer (oral form) in corrals with supplementary feeding.
This latter form was also observed in earlier times/ sources but has during the past few years become a problem in Fennoscandia, especially Finland and Sweden, where feeding is more common than in Norway, so far. Since this disease caused huge trouble in historical times, it is a disease that has left traces in the literature, and also has been delivered between generations as part of the traditional knowledge.
We have also provided a new short section, describing the observations conducted by herders today, when animals are affected by necrobacillosis (oral form). A more in-depth description of the disease will, to our opinion, be too detailed in pathological terms to be suitable for this manuscript and the journal, and it is thus referred to relevant descriptions for the interested reader
Round 2
Reviewer 2 Report
Dear authors.
Your changes has been accepted, even those you have not considered necessary to be changed (adequately justified).
Moreover, there are still several foot notes that should be removed from the paper.
Author Response
Dear reviewer 2
We are glad that you are satisfied with our changes.
We have gone through the footnotes one more time. We agree with you that all footnotes are not necessary. So we have made these changes:
Out of 22 footnotes we have deleted 2, we have moved 7 into the main text while 13 remain unchanged.
The deleted ones were unnecessary repetitions, the ones moved into the text where relatively short and suited rather well there, while the remainder would have become challenges to text fluency.
Reviewer 3 Report
The authors did a good job in reviewing the article and responded to all points raised by me and other reviewers. However, I personally still believe that the type of article and type of analysis fits more into another type of journal, with a more sociological than ecological appeal.Author Response
Dear reviewer 3
We are glad that you are satisfied with our changes.
As for the question of this contribution's suitability for this journal, we do not comment upon that, but leave the issue as a decision of the editors.